# Water level change of Lake Machang in eastern China over 1814–1902 AD

Jie Fei[1]

Institute of Historical Geography, Fudan University, Shanghai 200433, China

## Abstract

Lake Machang, occupying an area of approximately 30 km$^2$ in Jining City, eastern China, was a historical reservoir on the Grand Canal existing from early 15th century to early 20th century. The premodern monthly water level observation of Lake Machang commenced in 1814 AD and ended in 1902 AD. The available observation data from the monthly records covered 75.6% of the entire study period 1814-1902. Although the water level was seemingly artificially intervened by human activities, monthly and annual water level changes still correlated well with precipitation. That is, climate is still the dominant factor of water level changes on seasonal and annual scales. The flooding of the Yellow River in 1871 carried a large amount of silt into Lake Machang, which resulted in the rise of lake bed and reclamation initiated by local residents. After the reclamation activity was officially approved in 1900, Lake Machang was massively reclaimed and eventually dried up in the early 20th century.

## Key words

Lake Machang, reservoir, Grand Canal, reclamation, water level

---

[1]  Email: jiefei@fudan.edu.cn

## Introduction

Historical reservoir evolution is a promising subfield of climatic change studies (Cardoso-Silva et al., 2021; Margarint et al., 2021; Bábek et al., 2021; Fei et al., 2021; Halac et al., 2020). However, water level change of reservoirs needs to be interpreted carefully, as is affected by a combination of factors. Historical textual records are effective in studying the long-term evolution of lakes, but they are fragmentary and qualitative, thus hampering the reconstruction of water level change with high resolution. Here we utilize a combination of premodern monthly water level observation data, textual records, and historical maps to reconstruct the evolution history of Lake Machang in Jining City, eastern China, and differentiate the effects of various factors, including climate, hydrology and human activities. To improve the practice of using water level data to understand past climate change, the water level change of this reservoir during 1814-1902 AD has been reconstructed in this work. (Figure 1).

Lake Machang, which occupies an area of nearly 30 km$^2$, was a historical reservoir on the Grand Canal (Figure 1) that had existed for several centuries before it dried up in the early 20th century. The climate in this area is a warm temperate semi-humid east Asian monsoon type. The monthly average temperature varies from $-2$ °C in January to 27 °C in July. The annual precipitation is around 700 mm and mainly occurs in summer as monsoon precipitation (Shen et al., 2008).

The Grand Canal, stretching around 1,800 km, is a world heritage site. Running from Beijing in the north to Hangzhou in the south, it is one of the greatest artificial waterways constructed in historical times in the world. Constructed in sections from the 5th century BC onwards, the current waterway system was completed in the late 13$^{th}$ century (Ji, 2008).

The middle section of the Grand Canal was repaired and modified in 1411. A group of reservoirs were established along the canal to ensure the water supply (Zhu, 2014; Fei et al., 2021). Water was collected in the reservoirs in every autumn when the monsoon precipitation was over and supplied the canal in spring until the monsoon precipitation came (Lu, 1775).

The channels of River Guang and a few small rivers were also slightly modified, and water was introduced into the Machang area and forming a new reservoir, which was named as Lake Machang (Yang, 1430; Figure 1). The official gazetteer recorded that the area was previously a horse pastureland; therefore, the new *shuigui* (reservoir) was named Machang Hu[2] (Xu, 1859). In this regard, the official documents indicated that Lake Machang formed in 1411.

However, a poem indicated that a lake already existed in this area by early 14th century. The poem is entitled *West Lake of Jizhou Prefecture*[3].… The author LI Gang

---

[2] *Ma* means horse, *Chang* means pastureland, and *Hu* means lake.

[3] The title of the poem is *Jizhou Xihu* (濟州西湖) in Chinese. Jizhou Prefecture was the historical name of Jining City during the Yuan dynasty (1271-1368). The original Chinese text reads, 渺渺澄湖望不窮，畫船曾駐夕陽中。 *miao miao cheng hu wang bu qiong, hua chuan ceng zhu xi yang Zhong*. The English translation is as follows, the lake is clean

was the mayor of Jizhou Prefecture in the period of 1324–1327. Therefore, the poem was probably created in the early 14[th] century.

**Materials and results**

Water level observations of reservoirs along the canal were organized by the General Administration of the Grand Canal[4] since the middle 18th century (Fei, 2009; Fei et al., 2012; Fei et al., 2021). The extant water level observation records of Lake Machang date back to 1763; however, early observations are fragmental and insufficient to establish a chronology (Academy of Water Conservancy and Hydroelectric Power, 1988).

In 1814, the emperor decided to further regulate the water supply and ensure the canal transportation. He ordered that the water levels of the reservoirs along the Grand Canal should be observed monthly and the observation reports should be directly submitted to the emperor himself (Academy of Water Conservancy and Hydroelectric Power, 1988). Therefore, the water level observations were trustworthy and very reliable.

Monthly observation of the water level of Lake Machang as well as other reservoirs along the Grand Canal were henceforth organized since 1814. However, the function of these reservoirs ended in 1902, when the General Administration of the Grand Canal was dissolved (Fei et al., 2021). The observations of the water levels of the reservoirs along the canal, including Lake Machang, were therefore terminated in 1902.

The extant observation data of Lake Machang could cover 75.6% of the entire study period 1814–1902. The missing points, which account 24.4% over 1814–1902, were interpolated using the mean of two neighboring points. The observations followed the Chinese lunar calendar months, and they were conducted at the end of every month. A unique length unit *yingzao chi* (1 *yingzao chi* = 0.32 m) was adopted in the observations (Table 1). *Yingzao chi* was an official length unit during the Qing Dynasty. It was widely adopted in hydraulic engineering and relevant affairs (Wanyan, 1836). Notably, the water levels were not those of above sea level but the water depths at the observation station. A water level ruler was erected somewhere on the bank of Lake Machang. However, no relics or records of the water level ruler of Lake Machang are available to date. The original water level observation reports are scattered through the imperial archives of the Qing Dynasty (1644–1912), which are documented in the First Historical Archives of China[5]. After converting the observation data into SI unit and AD dates, the chronologies of the annual mean, maximum, and minimum water levels were established (Figure 2). Accordingly, the average water level of Lake Machang over 1814-1902 was 0.92 m. In other words, the average depth of Lake Machang was

and vast, and I cannot see the shoreline.

[4] The General Administration of the Grand Canal (*Hedao Zongdu Yamen*) was an official department of the central government of the Qing Dynasty (1644-1912). It was located in Jining City, and was responsible for the transportation and water supply of the Grand Canal, as well as the water level observation of the reservoirs along the Grand Canal.

[5] The First Historical Archives of China (*Zhongguo Diyi Lishi Dangan Guan*) is an official department of the Chinese central government. It is a national archive of China and collects the archives of the Ming (1368-1644) and Qing (1644-1912) dynasties.

less than 1 m, therefore it was really a shallow water reservoir, and vulnerable to environmental change.

**Comparison with relevant precipitation chronologies**

The water level variability of Lake Machang was compared with that of precipitation on monthly and annual scales.

The average monthly water level variability of Lake Machang in the period of 1814–1902 was compared with the average monthly precipitation variability of Jining City in modern times (1951–2000. Figure 5). We calculated the correlation (R) between the two variables, and found that the monthly water level responded well with precipitation but with a time-lag of 2 months (R=0.753, N=12). As we mentioned above, the water level of Lake Machang as a reservoir was artificially intervened in order to ensure the water supply of the Grand Canal. Water from the drainage basin was collected in summer and autumn (rainy season of this area). The transportation of the Grand Canal usually paused in winter, as the channels were frozen. The transportation usually restarted in February or March when spring came. As precipitation was low in spring in this area (Figure 5), water collected in Lake Machang as well as other reservoirs was discharged into the Grand Canal to ensure the water supply of transportation (Academy of Water Conservancy and Hydroelectric Power, 1988). This process possibly explained the time-lag of two months of monthly water level variability.

The annual water level variability of Lake Machang was compared with the Dryness Wetness Index (hereinafter DWI) dataset described in the Central Meteorological Administration of China (1981). This dataset is based on the textual records on precipitation in the historical local gazetteers in China. It covers 120 stations, including four stations in the vicinity of Lake Machang, namely, Heze, Jinan, Linyi, and Xuzhou (Figure 1). DWI is a five-grade dataset, i.e., 5 (very dry), 4 (dry), 3 (normal), 2 (wet), and 1 (very wet). We calculated the correlation of the average DWI of Heze, Jinan, Linyi, and Xuzhou ($DWI_{HJLX}$) with the annual mean, maximum, minimum water levels of Lake Machang in the period of 1814–1902, and the correlation coefficients (R) of $R_{mean} = -0.50$, $R_{max} = -0.52$, $R_{min} = -0.41$ (N=89). All these values are significant. Furthermore, the relatively high correlation value indicates that precipitation was a crucial factor of the annual water level changes of Lake Machang in the period of 1814–1902.

We further examined the ten years with highest water levels and another ten years with lowest water levels. These years with highest or lowest water levels will be compared with the historical records of local flood and drought. The ten years with highest annual maximum water levels are 1898, 1820, 1852, 1860, 1883, 1864, 1863, 1819, 1839, and 1892. Among them, all but two years (1883, and 1864) corresponded with records of local floods. The ten years with lowest annual minimum water levels are 1901, 1902, 1814, 1857 1874, 1850, 1866, 1847, 1837, and 1856. Among them, only four years (1901, 1814, 1874, and 1856) corresponded with records of droughts.

The comparison possibly indicated that the extreme value of water level did not link closely with local disasters, no matter flood or drought. Furthermore, the droughts seldom resulted in the drying up of Lake Machang in the period of 1814–1902, and

only led to abnormally low water levels in winter and spring. The lake usually recovered in several months when summer monsoon came. This proved that precipitation affected annual maximum water level more significantly than annual minimum water level.

Beijing lies approximately 490 km north of Lake Machang, and the correlation coefficient of the annual precipitation of Beijing and Jining over the period of 1951–2010 is 0.148 (N=60). Beijing has the longest premodern and modern meteorological observation histories in China. Continuous modern meteorological observation in Beijing began in 1841. Premodern daily observations of precipitation days are available from 1724 (Beijing Meteorology Service, 1982). We established the chronology of annual precipitation of Beijing over the period of 1814–1902 using a combination of the above mentioned two types of sources (Figure 4). The correlation coefficient of the annual mean water level of Lake Machang and the annual precipitation of Beijing over the period of 1814–1902 is merely 0.021 (N=89). This indicated that the water level of Lake Machang was not a large-scale climate indicator, and it did not reflect the precipitation of a large area.

**Flooding of the Yellow River, silt sedimentation, and reclamation**

Wang et al., (1999) reconstructed the chronology of the runoff of the Yellow River at Sanmenxia City, using a combination of relevant historical records. It actually indicated the runoff of the upper and middle reaches of the Yellow River Basin. The correlation between the runoff of the Yellow River at Sanmenxia and the annual mean water level change of Lake Machang over 1814-1902 is merely 0.139 (N=89) (Figure 6). This indicated that the water level change of Lake Machang was not significantly affected by the runoff of the Yellow River.

Over the period 1814–1902, Lake Machang was only flooded by the Yellow River in 1851 and 1871, though the lake was only nearly 100 km away from the Yellow River, which flooded very frequently. The channel change of the Yellow River in 1855 was a major hydrological event in the history of China, but it did not directly affect Lake Machang.

The flooding of the Yellow River in 1851 was a large-scale hydrological disaster. It resulted in the southward migration of the Huaihe River (ca.300 km south of Lake Machang), which was also a major hydrological event in the history of China. Lake Nansi (ca.30 km southeast of Lake Machang) recorded an extremely high water level interval lasting four years over the period of 1851–1855 (Figure 7). However, Lake Machang was only moderately flooded by the Yellow River in 1851 (Figure 2).

The autumn of 1871 was very rainy and the Yellow River burst its banks at Yuncheng County, around 70 km to the northwest of Lake Machang (Cen, 1957). The breach was not filled up until the next spring. Notably, the flooding of the Yellow River in 1871 was also a large-scale hydrological disaster.

The flooding of 1871 did not result in extremely high water level in Lake Machang (Table 1). However, it carried a great amount of silt into the reservoir. The bed of the reservoir increased significantly due to the silt sedimentation carried by the floods of the Yellow River. The average water level of Lake Machang during 1814-1870 was 1.03 m, whereas that of 1871-1902 decreased to 0.72 m. From then on, the inflow of River

Guang no longer reached the reservoir. Local residents began to reclaim the reservoir (Pan, 1927).

On the other hand, the flooding of 1871 severely destroyed the banks of the Grand Canal in this region. There were four connected reservoirs along the Grand Canal to the south of Lake Machang before 1871. The dikes separating them were destroyed by the flooding of 1871, and these reservoirs merged into a united Lake Nansi (Fei, 2009; Fei et al., 2012; Fei et al., 2021).

The flooding of 1871 significantly affected the evolution of Lake Machang, and it marked the shrinkage of the reservoir and the beginning of the reclamation. The annual minimum water level of Lake Machang before and after 1871 were 0.70 m (1814-1870) and 0.39 m (1871-1902), respectively (Figure 2). Low water level could make reclamation easier and further accelerate the shrinkage of the reservoir. In 1900, the central government approved the local authority's application regarding the reclamation of Lake Machang. Two years later, the General Administration of the Grand Canal was dissolved, and the function of the Lake Machang as a reservoir of the Grand Canal was ended. Hereby, local residents poured in and massively reclaimed the reservoir.

As a result, Lake Machang gradually dried up in the following decades. The local authority organized a field investigation regarding the Grand Canal in Shandong Province in 1916 and drew a map entitled "*The Plan of the Southern Part of the Grand Canal, Including the Shallow Lakes and Swamps* (Scale 1:200,000) (Pan, 1916)" (Figure 8). Lake Machang was drawn as a dry lake in this map. In 1927, Lake Machang was also drawn as a dry lake in the local gazetteer of Jining (Yuan, 1927). From these maps, it could easily be concluded that Lake Machang dried up no later than 1916. Notably, the annual precipitation did not decrease significantly in the early 20th century (Central Meteorological Administration of China, 1981). Therefore, although the climate played a fundamental role in affecting the water level of Lake Machang, large-scale reclamation accelerated its drying up.

Overall, the road map of the drying up of Lake Machang was as follows: the flooding of 1871 carried a large amount of silt into the reservoir and therefore resulted in the rise of the lake bed and shrinkage of the reservoir, which caused the reclamation by local residents and further shrinkage of the reservoir. After the central government formally approved the reclamation activity in 1900, local residents poured in and further reclaimed it massively, and caused the dried up of Lake Machang in the early 20th century.

From the fate of Lake Machang, vulnerability of a local water body could come from both natural and human aspects. Under the current climate change and its natural impacts to water bodies, human adaption should be a key question in the era of Anthropocene.

**Comparison with Lake Nansi**

Lake Nansi lies 30 km southeast to the Lake Machang, and it is actually the general name of four connected reservoirs along the Grand Canal. The four reservoirs are Lake Nanyang, Lake Dushan, Lake Zhaoyang, and Lake Weishan. Water level observations were made for the four reservoirs. The average annual mean water level change of Lake

Nansi was calculated and compared with that Lake Machang over the period of 1814–1902. The correlation coefficient is 0.374 (N=89) (Figure 7). The annual water level change of Lake Machang showed great similarity to those of its neighbor reservoirs.

On the contrary, the long-term evolution of Lake Machang and Lake Nansi were very different. Lake Machang was reclaimed and dried up in the early 20th century, but Lake Nansi gradually expanded (Fei et al., 2021). Lake Nansi was even more frequently flooded by the Yellow River. For example, the flooding of 1871 destroyed the dikes separating these reservoirs, thus forming a united Lake Nansi (Fei, 2009; Fei et al., 2012; Fei et al., 2021).

From the perspective of geomorphology, the altitude of Lake Machang is a little higher than that of Lake Nansi, and the Grand Canal in this region flows southeastward. That is, water flew from Lake Machang to Lake Nansi along the Grand Canal. When Lake Machang was reclaimed, water that could otherwise be collected in it directly flew into Lake Nansi, and resulted the expansion of Lake Nansi. Geologically, the basin of Lake Nansi is slowly subsiding, whereas that of Lake Machang is stable (Shen et al., 2008). The subsiding possibly compensated the silt sedimentation in Lake Nansi, whereas Lake Machang was silted up and reclaimed.

**Conclusions**

We reconstructed the water level change of Lake Machang over the period of 1814–1902 and the evolution history by using premodern monthly water level observations and other historical records. Precipitation was still a dominant factor of water level change of Lake Machang on monthly and annual scales, though human activities intervened the monthly water level change.

The flooding of the Yellow River in 1871 carried a great amount of silt into Lake Machang. The central government formally approved the reclamation activity of Lake Machang in 1900. The administration of the Grand Canal was dissolved two years later, and the function of the Lake Machang as a reservoir of the Grand Canal was ended. Local residents poured in and massively reclaimed Lake Machang, and resulted in the dried up in the early 20th century.

Shallow lakes and reservoirs are vulnerable to climatic and environmental changes, and human activities like reclamation could accelerate the drying up of water bodies.

**Supplement**. The supplement related to this article is available online at: …
**Competing interests**. The authors declare that they have no conflict of interest.

**Acknowledgments** This research was supported by the Shanghai Municipal Philosophy and Social Science Grant (No. 2017BLS003).

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

Table 1. Water level observations of Lake Machang in a Chinese lunar calendar year (The 10th year

of the Tongzhi Reign Period, that is, from 19 Feb 1871 to 8 Feb 1872).

| Observing dates in Chinese lunar calendar | Observing dates in AD | Water levels in *yingzao chi** | Water levels in SI unit (m) |
|---|---|---|---|
| 30th, 1st month** | 20 Mar 1871 | 1.2 | 0.384 |
| 30th, 2nd month | 19 Apr 1871 | 1.2 | 0.384 |
| 29th, 3rd month | 18 May 1871 | 1.2 | 0.384 |
| 30th, 4th month | 17 Jun 1871 | 1.4 | 0.448 |
| 30th, 5th month | 17 Jul 1871 | 1.6 | 0.512 |
| 29th, 6th month | 15 Aug 1871 | 1.8 | 0.576 |
| 30th, 7th month | 14 Sept 1871 | 1.7 | 0.544 |
| 29th, 8th month | 13 Oct 1871 | 2.1 | 0.672 |
| 30th, 9th month | 12 Nov 1871 | 2.1 | 0.672 |
| 29th, 10th month | 11 Dec 1871 | 2.1 | 0.672 |
| 29th, 11th month | 9 Jan 1872 | 1.9 | 0.608 |
| 30th, 12th month | 8 Feb 1872 | 1.9 | 0.608 |

* denotes water levels in the length unit *yingzao chi* (1 *yingzao chi* = 0.32 m).

** denotes the 30th day (i.e., the month end) of the 1st month.

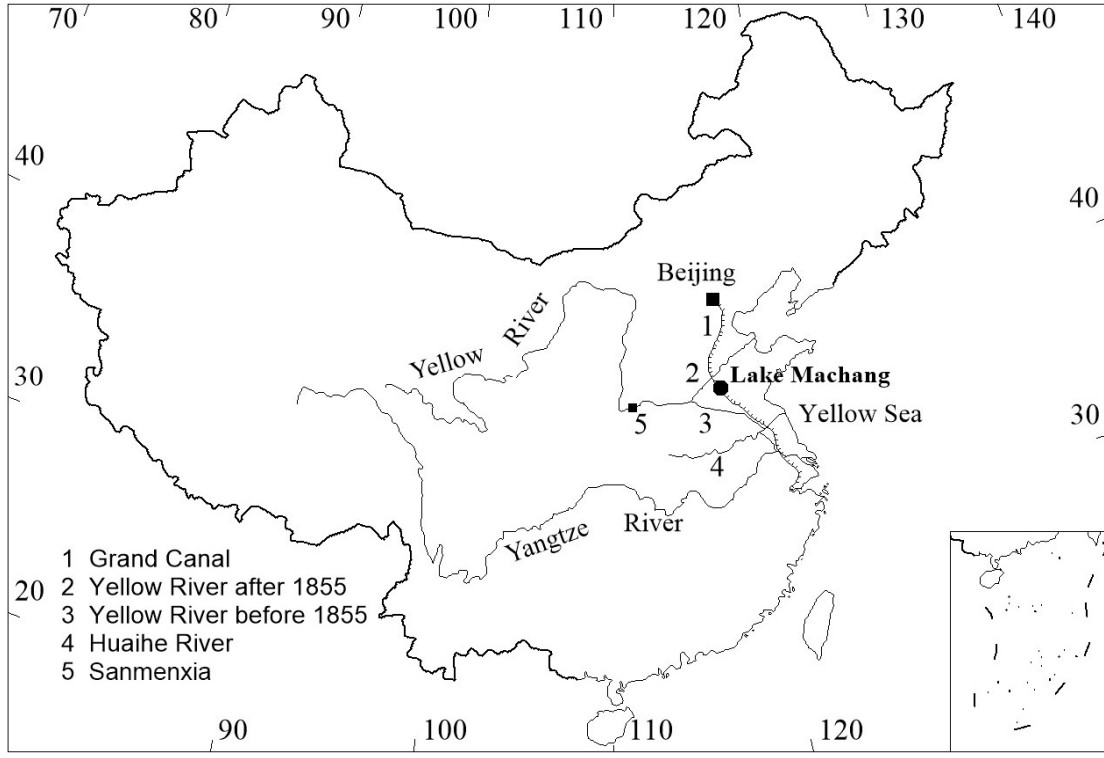

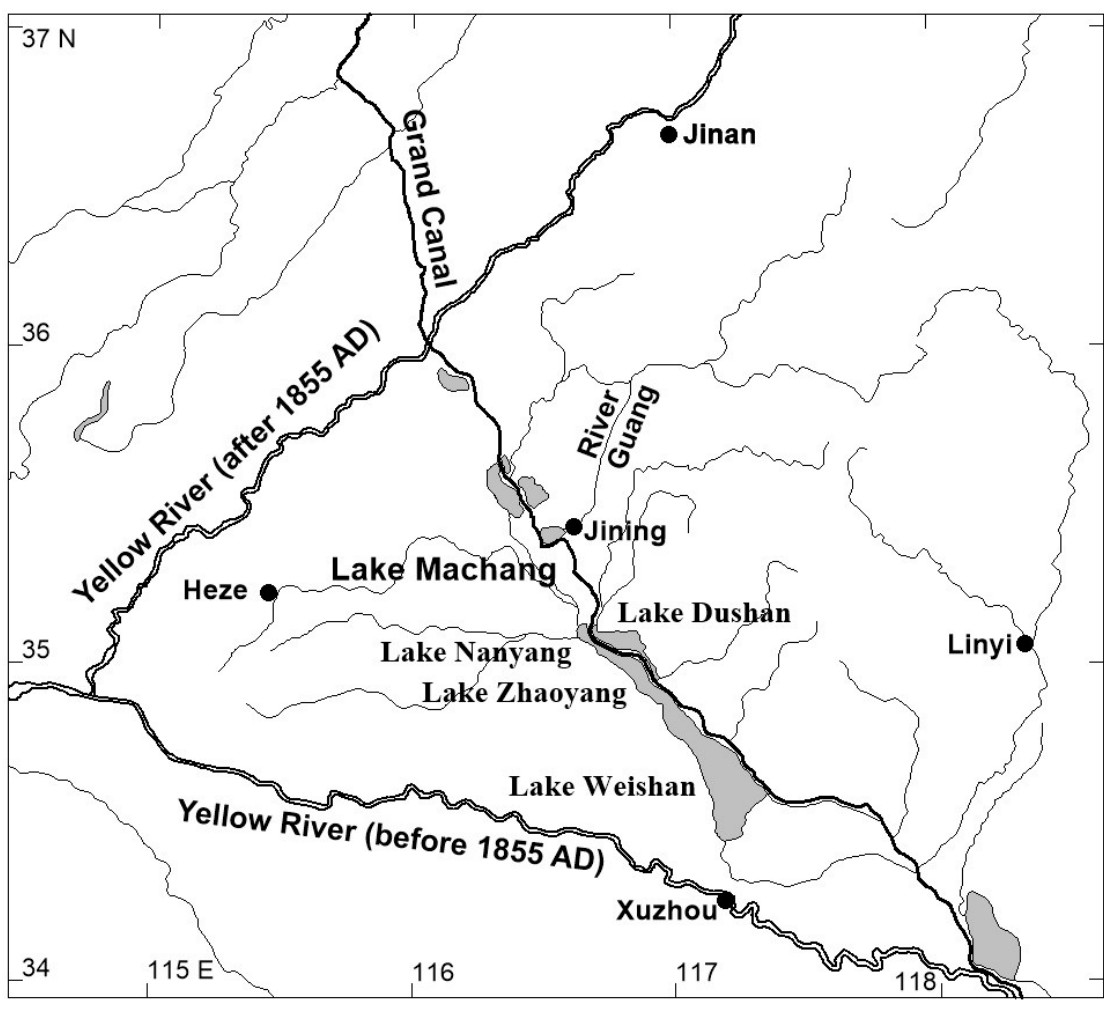

Figure 1. Maps showing the location (upper part) and vicinity (lower part) of Lake Machang.

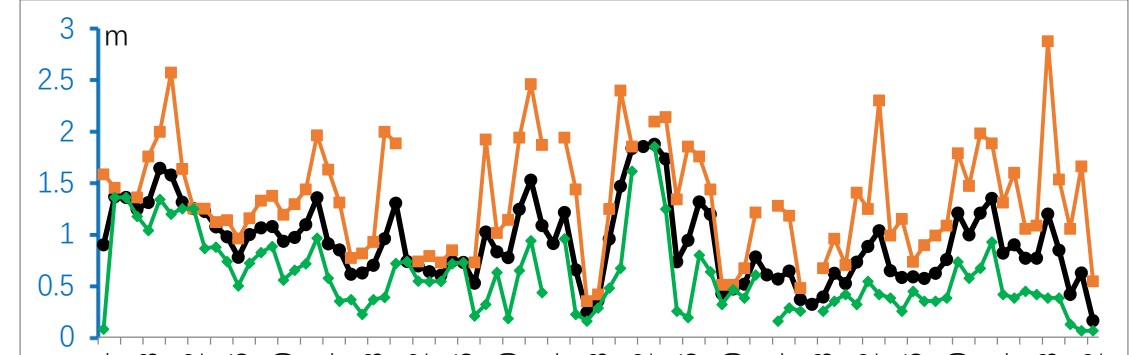

Figure 2. Annual mean (thick black line with dots), maximum (thin brown line with squares), and
minimum (thin green line with diamonds) water levels of Lake Machang over the period of
1814–1902.

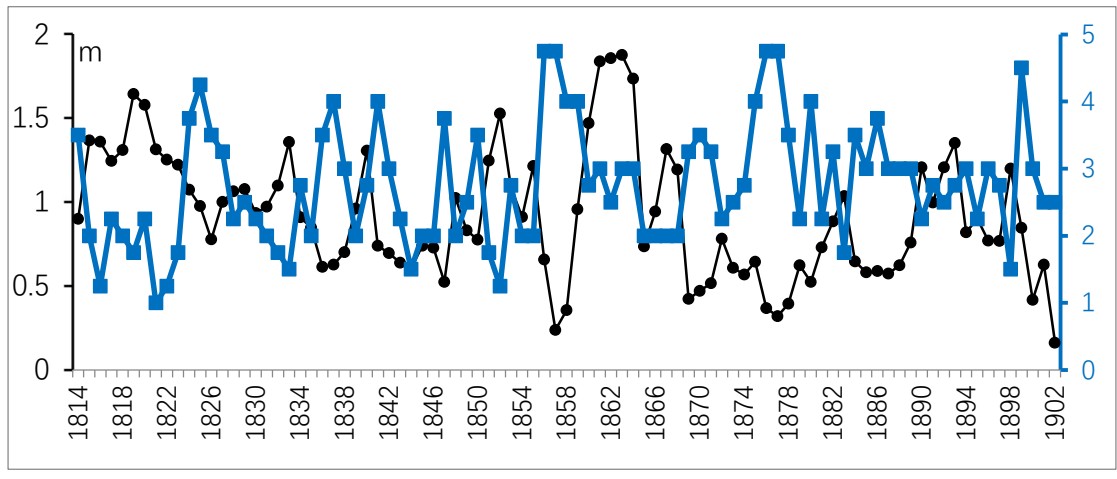

Figure 3. The thick blue line with squares denotes the Average Dryness Wetness Index (hereinafter

DWI) of four stations in the vicinity of Lake Machang, namely, Heze, Jinan, Linyi, and Xuzhou.

DWI is a five-grade dataset, that is, 5 (very dry), 4 (dry), 3 (normal), 2 (wet), and 1 (very wet).

The thin black line with dots denotes the annual water level of Lake Machang.

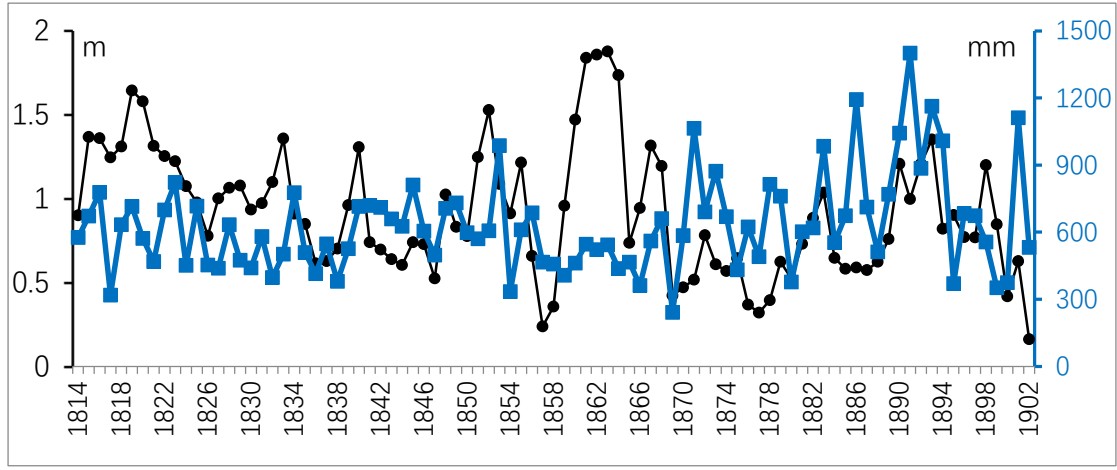

Figure 4. Annual precipitation of Beijing over the period of 1814–1902 (thick blue line with squares)

and its comparison with annual mean water level of Lake Machang (thin black line with dots).

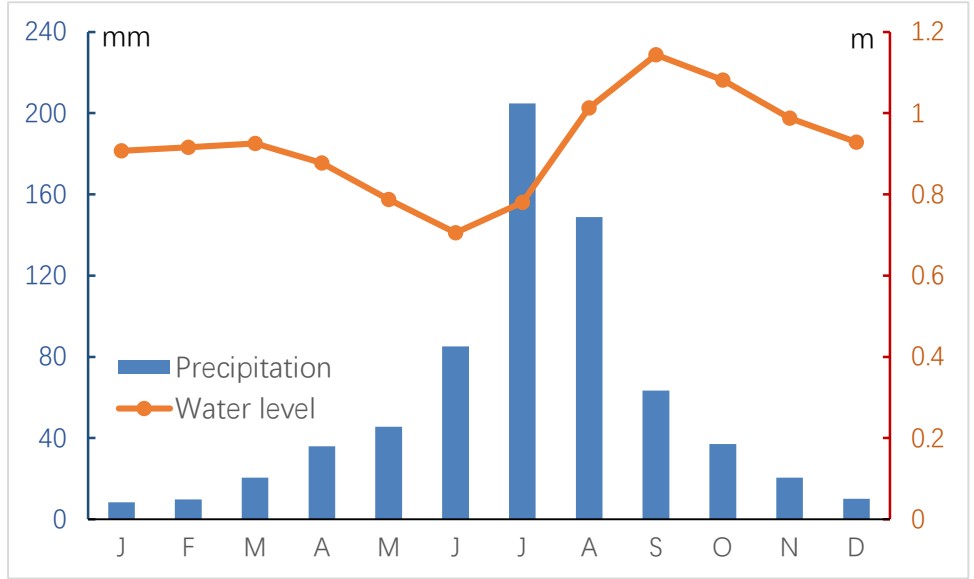

Figure 5. Comparison of the average monthly water level variability of Lake Machang (1814–1902)
with the monthly precipitation variability of Jining City (1951–2010).

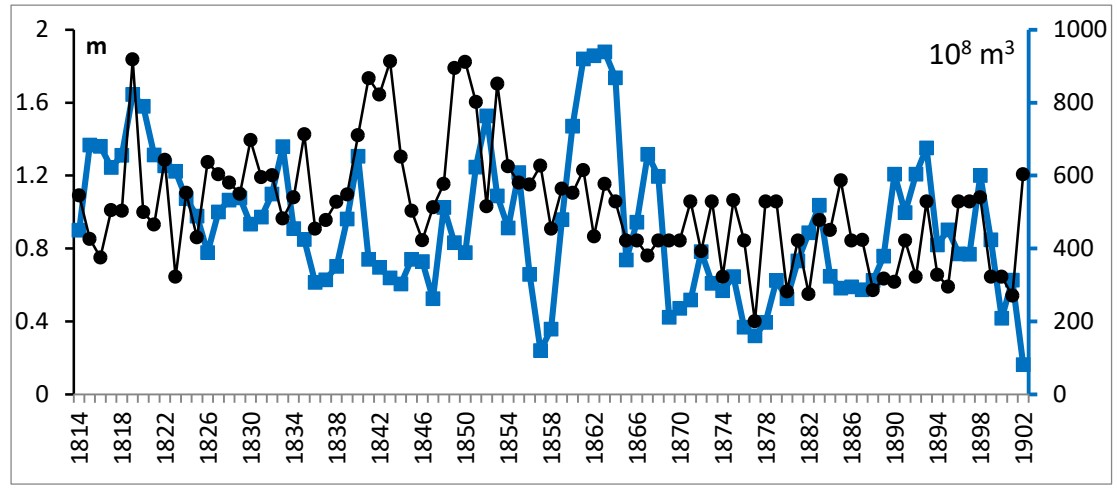

Figure 6. Runoff of the Yellow River at Sanmenxia (thick blue line with squares) over 1814-1902
and its comparison with annual mean water level of Lake Machang (thin black line with dots).

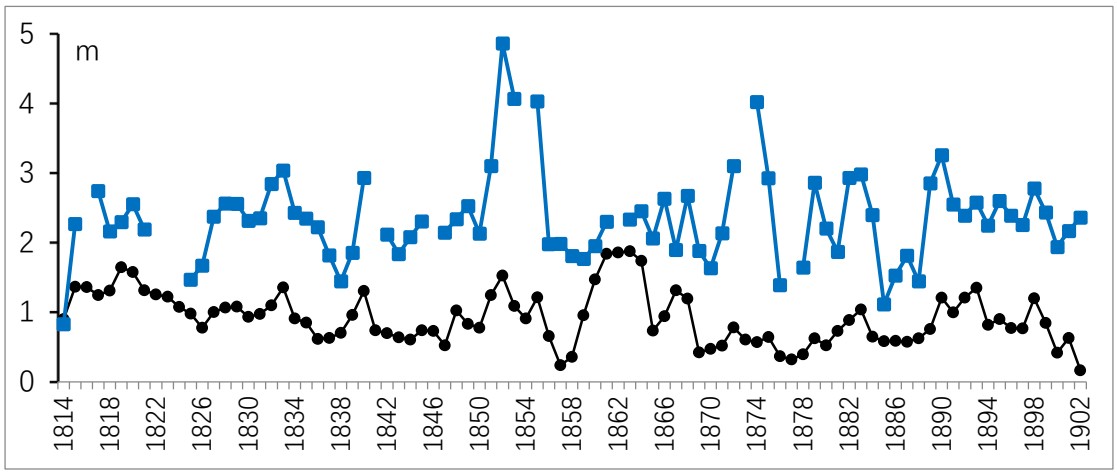

Figure 7. Annual mean water level change of Lake Nansi (thick blue line with squares) over 1814-
1902 and its comparison with annual mean water level of Lake Machang (thin black line with dots).

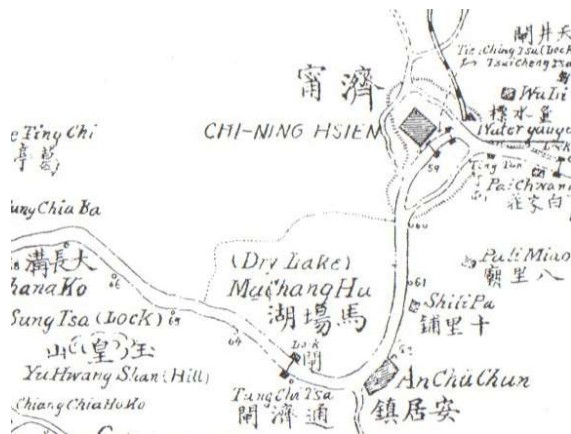

Figure 8. Part of the historical map entitled "*The Plan of the Southern Part of the Grand Canal,*
*including the Shallow Lakes and Swamps* (Pan, 1916. See Figure S1 for the whole map)"
showing the area near Lake Machang. Lake Machang was noted as a 'dry lake' and'*Machang*
*Hu*' (Hu means lake).

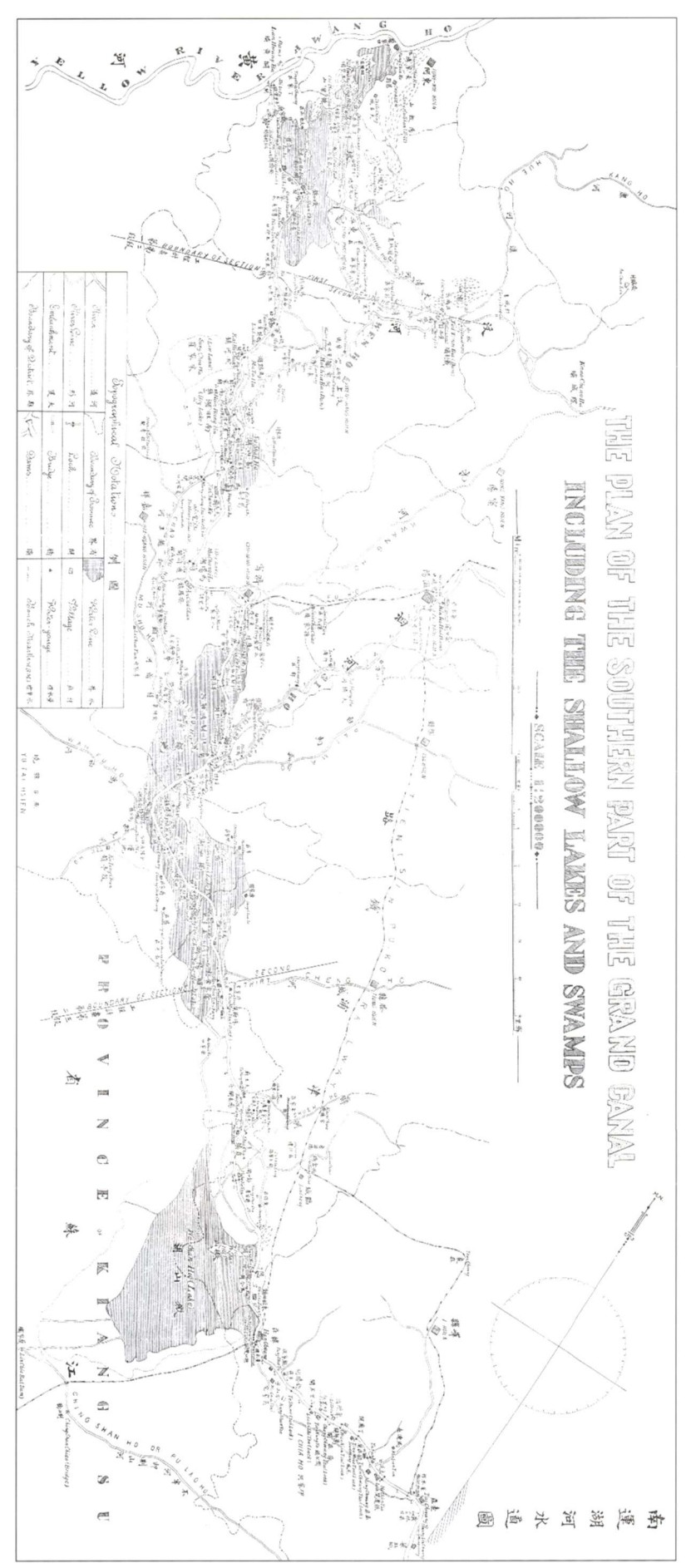

THE PLAN OF THE SOUTHERN PART OF THE GRAND CANAL
INCLUDING THE SHALLOW LAKES AND SWAMPS

SCALE 1:300000

Figure S1. the historical map entitled "*The Plan of the Southern Part of the Grand Canal, including*
*the Shallow Lakes and Swamps* (whole map. Pan, 1916)