# Peer review of "Water level change of Lake Machang in eastern China"

_Climate of the Past, 2021_

## Author Response (AR1)

Dear Colleague,

Thank you and the kind reviewers very much for all the constructive and helpful comments and suggestions. I accepted and responded to all the comments and suggestions one by one, and revised the manuscript carefully.

The language is polished.

My only concern is about the historical map in Figure 8. It is actually a copy of a historical map, and the original quality is not very high. I feel difficult to improve the quality, and wonder whether it is possible to remove it in the text and attach it as a Supplementary Materials.

With Best Regards
Jie

**Response to Reviewer 1**

Dear Colleague,

Thanks a lot for your kind and constructive comments. All of them are accepted and responded one by one. Please see the following for the details.

With Best Regards

Jie

**Line 36: "Historical reservoir evolution is a promising subfield of climatic change studies." This is an eye-catching statement. But, the findings in this study show that the historical reservoir (i.e., Lake Machang) is subject to the strong influence of human activities. To what extent could historical reservoirs reflect climate change?**

RE: Thanks. A sentence is added in the first paragraph of 'Introduction' Section, i.e., 'The water level change of reservoirs needs to be interpreted carefully, as is affected by a combination of factors.' Please see lines 37-38.

**Lines 100–103: "The average monthly water level variability of Lake Machang in the period of 1814–100 1902 AD was compared with that of Jining City in the period of 1951–2000 AD (Figure 5). We found that the monthly water level responded well with precipitation but with a time-lag of 2 months." Please show how to calculate the time-lag.**

RE: the sentence is modified as following,

We calculated the correlation (R) between the two variants, and found that the monthly water level responded well with precipitation but with a time-lag of 2 months (R=0.753, N=12). Please see lines 117-119.

**Lines 135–146: This paragraph shows that the changes in the water level in Lake Machang are site-specific in nature. So, how to link the findings in this study to climate change, which is a macro-regional phenomenon? Besides, apart from precipitation, would the changes in water level in Lake Machang be caused by other natural factors such as the changes in temperature, monsoon, or ocean/atmospheric circulation?**

RE: a paragraph is added comparing the water level with the runoff of the Yellow River. Please see lines 170-176.

Wang et al., (1999) reconstructed the chronology of the runoff of the Yellow River at Sanmenxia, using a combination of relevant historical records. It actually indicates the runoff of the upper and middle reaches of the Yellow River Basin. The correlation between the runoff of the Yellow River at Sanmenxia and the annual mean water level change of Lake Machang over 1814-1902 was merely 0.139 (N=89). This indicated that the water level change of Lake Machang was not significantly affected by the runoff of the Yellow River.

**Lines 233–236: Perhaps it could be more specific in stating what people could learn from the history of Lake Machang.**

RE: a sentence is added in the end of the Conclusion Section stating what people could learn from the history of Lake Machang.

'shallow lakes and reservoirs are vulnerable to climatic and environmental changes, and human activities like reclamation could accelerate the dry up of such water bodies.'

Please see lines 252-253.

**Figure 2: The three panels could be combined into one.**

RE: Yes. The panels are combined into one.

**Figures 3, 4, and 6: The annual mean water levels of Lake Machang in 1814–1902 could be put into the figure for comparison.**

RE: Yes, the chronology is added in the relevant figures for comparison.

Response to Reviewer 2

Dear Colleague,

   Thanks a lot for your kind and constructive comments. All of them are accepted and responded one by one, except the last issue. Please see the following for the details.

   With Best Regards

Jie

**The title of the paper is misleading as not the last 200 years play a significant role in this paper but rather the focus is on the period 1814 to 1902, for which records exist, after that the lake dried up and was reclaimed. This focus should be reflected in the paper's title.**

RE: The titled is modified as, Water level change of Lake Machang in eastern China over 1814–1902 AD

**In the introduction "1814 to 1912" is mentioned as the study period but the period of 1902 to 1912 is not very much discussed. Is this a typo, did you mean 1902 here? If not, I would suggest to elaborate on the last decade of the study period.**

RE: Sorry. It is a typo. Please see line 13 for the correction.

**It is difficult to establish the role of climate change on this lake as the water availability was heavily influenced by human actions. These human actions manipulating the water supply of the lake and the Grand Canal play a significant part in this story, could you perhaps detail the "human actions" more precisely, how was water supply controlled exactly? Can you give examples?**

RE: Thanks. A sentence is added in the first paragraph of 'Introduction' Section, i.e., 'The water level change of reservoirs needs to be interpretated carefully, as is affected by a combination of factors.' Please see lines 37-38.

The details of human actions were significantly strengthened. Please lines 120-128.

**As this special issue analyzes archives of society, a more critical approach towards the historical sources used for this paper would be desirable, i.e. assessing how reliable the sources are, which time period were they produced in and by whom, what makes them reliable?**

RE: thanks for the suggestion. A paragraph illustrating this issue is added in the Materials and Results Section. Please see lines 82-87.

**I think it would help the paper if a more precise research question was formulated in the introduction.**

RE: thanks. The first paragraph of the Introduction Section is rewritten, and a precise research question is added in the end of this paragraph.

Please see lines 41-44.

**The paper would benefit from language editing, particularly regarding articles and prepositions.**

RE: Yes. The language of the manuscript is polished. Please note that such modifications are not highlighted in the text.

**I feel it is not necessary to add "AD" to every year mentioned as it becomes clear that this paper focuses on the 19th and 20th centuries in the Common Era.**

RE: "AD" is removed except the first appearance in the Abstract and Introduction sections.

**Some specific remarks:**

**Lines 45-47 and/or lines 69-70: For the international audience that Climate of the Past aims at and that is probably not familiar with the history of the Grand Canal, I suggest introducing the Grand Canal with slightly more background information. Something along the lines: When was it created, where did it begin and end, how long was it?**

RE: Thanks for the suggestion. The introduction of the Gran Canal is significantly strengthened and moved to the 3$^{rd}$ paragraph of the Introduction Section.

The Grand Canal, running from Beijing in the north to Hangzhou in the south, is a world heritage site. Stretching 1,794 kilometers, it is one of the greatest artificial waterways constructed in historical times in the world. Constructed in sections from the 5th century BC onwards, the current waterway system was completed in the late 13th century (Ji, 2008).

Please see lines 54-58.

**Line 60: Thank you for including the Chinese original in the footnotes and for explaining the meaning of the lake's name here. This was very helpful.**

RE: Thanks.

**Lines 62-66: This historical background into the creation of the lake is interesting. The translated quotations from the poem should be but in quotation marks. I do not understand how you conclude that the poem must have been written during Li Gang's time as mayor, couldn't he have written it before or after?**

RE: thanks for the suggestion. Translation of the poem is moved to the footnote.

Early 14$^{th}$ century would be more reasonable than '1324–1327'.

The sentence was modified as following,

'Therefore, the date of this poem should be early 14th century.' (lines 72-73)

**Lines 77-82: Can you tell us more about the role, tasks, and responsibilities of the Grand Canal administration? Were they responsible for the observations and the regulations of the water supply?**

RE: Thanks. A footnote is added as following,

The General Administration of the Grand Canal (Hedao Zongdu Yamen) was an official department of the central government of the Qing Dynasty (1644-1912 AD). It was located in Jining City, and was responsible for the transportation and water supply of the Grand Canal, as well as the water level observation of the reservoirs along the Grand Canal.

Please see Footnote 4, page 3.

**Lines 83-85: Are there gaps in the data between 1814 to 1902 that make up 24.4% or do you refer to gaps in the data and the ten years following 1902?**

RE: actually I mean the gaps between 1814 and 1902 make up 24.4%.

In order to avoid misunderstanding, the sentence is modified as follows, 'The missing points, which account 24.4% over 1814–1902.' Please see lines 93-94.

**Lines 87-89: You write that the observations of the water levels did not relate to the sea level but to the observation station, which has not survived. Can you elaborate here on how you deciphered the water levels in the lake from the records? By saying yingzao chi was a unique length unit, do you mean it was uniquely used at this location or is it unique in the sense that it specifically refers to 0.32 meters?**

RE: the explanation of the details of observation was strengthened in 4[th] paragraph of the Material and Results Section. Please see lines 99-101.

A sentence is added illustrating the length unit, and a reference is also added in the reference list. Please see lines 98-99 and 287-288.

'Yingzao chi was an official length unit of the Qing Dynasty. It was widely adopted in hydraulic engineering and relevant affairs.'

'Wanyan Linqing, 1836. *Hegong Qiju Tushuo* (Illustrated Handbook of hydraulic engineering). Published in 2015 by the Zhejiang people's fine arts press.'

**Lines 90-93: Are the First Historical Archives of China an archive or a critical edition? Is this a reliable historical source, if so, what makes it a reliable source?**

RE: thanks for the suggestion. A footnote is added and it demonstrates what is the first historical archives. The footnote is as following,

The First Historical Archives of China (Zhongguo Diyi Lishi Dangan Guan) is an official department of the Chinese central government. It is one of the national archives of China and collects the archives of the Ming (1368-1644 AD) and Qing (1644-1912 AD) dynasties.

Please see Footnote 4, page 3.

**Lines 162-166: Do we know how deep Lake Machang was before and after the deposition of silt in the lake?**

RE: we calculated the average water level of Lake Manchang, and two sentences are added in the end of the Materials and Results Section. Please see lines 98-100.

'Accordingly, the average water level of Lake Machang over 1814-1902 was 0.92 m. In other words, the average depth of Lake Machang was less than 1 m, therefore it was really a shallow water reservoir, and vulnerable to environmental change.'

**Lines 213-220: Thank you for including some geomorphological backgrounds here, this was helpful.**

RE: Thanks.

**Lines 233-236: It is unclear how we can relate from a former 30 km2 lake at which the drought and flooding conditions in the region could be mitigated by human action to the world at large in the face of anthropogenic climate change. I suggest elaborating on the matter or deleting this paragraph.**

RE: thanks. The paragraph is slightly modified and moved to the end of the Section 'Flooding of the Yellow River, silt sedimentation, and reclamation'. Please see lines 231-234.

Instead, a new sentence is added here as following,

'Shallow lakes and reservoirs are vulnerable to climatic and environmental changes, and human activities like reclamation could accelerate the dry up of such water bodies.'

Please see lines 271-272.

**Figures: The maps are very helpful to give an international audience a general idea of where the Grand Canal and Lake Machang are located.**

**Figure 1, top map: The location of the Bohai Sea seems irrelevant for the paper, if this is correct, please delete "5 Bohai Sea" from the map, it is confusing.**

RE: the Bohai Sea is removed. Instead, Sanmenxia is added as a new chronology is added in the text.

**Figure 1, bottom map: I suggest editing this map to show more clearly which lake is Lake Machang, simply by adding a line or arrow from the name "Lake Machang" to the respective lake. It might also aid the reader to indicate which lake(s) is Lake Nansi. Please also add an indicator for the scale of the map (1 cm = how many kilometers).**

RE : the figure is revised accordingly. However, we feel difficult to add an accurate scale. Anyway, the figure has grids of longitude and latitude.

**Figure 7: This map needs to be provided in better quality and please indicate in the larger map where the outcrop showing Lake Machang is located.**

RE: this is actually a historical map, and the quality cannot be improved. The scale is big and it needs an entire page to show the details, so it would be better to remove it from the text, and just use the enlarged part. Is it possible to attach the entire map as a supplementary material please?